# Influence of Simulated Altitude Exposure (2500 m) on Patients with Fontan Palliation Based on Circulating Hypoxia-Associated Factors

**DOI:** 10.3390/ijms26030887

**Published:** 2025-01-21

**Authors:** Nicole Müller, Christopher Hart, Julian Alexander Härtel, Jens Jordan, Jens Tank, Johannes Breuer, Marijke Grau, Stilla Frede, Frank Splettstoesser, Tobias Kratz

**Affiliations:** 1Department of Pediatric Cardiology, University Hospital Bonn, 53127 Bonn, Germany; christopher.hart@ukbonn.de (C.H.); j.haertel@paed-sportmedizin.de (J.A.H.); johannes.breuer@ukbonn.de (J.B.);; 2Department of Molecular and Cellular Sports Medicine, German Sport University Cologne, 50933 Cologne, Germany; 3Institute of Aerospace Medicine, German Aerospace Centre (DLR), 51147 Cologne, Germany; 4Department of Anaesthesiology, University Hospital Bonn, 53127 Bonn, Germany

**Keywords:** Fontan, hypoxia-induced factor, altitude, univentricular heart

## Abstract

Patients with a univentricular heart live with chronic hypoxia (75–85%) in their first years of life, which could affect adaptation to altitude or other hypoxic insults later in life. To test this hypothesis, we exposed 18 patients with Fontan circulation (age: 24.5 [16.3–38.8] years; f/m 9/9) to simulated altitude using normobaric hypoxia (15.2% oxygen, equivalent to 2500 masl) for 24 h. In blood samples obtained in normoxia (T1, 21% oxygen) and after 24 h hypoxia after a submaximal stress test, we measured hypoxia-regulated molecules involved in angiogenesis and tissue homeostasis. A significant increase was displayed for IL-10 (*p* = 0.001), CCL2 (*p* = 0.006), ANG-1 (*p* = 0.001), ANG-2 (*p* = 0.029), FGF-1 (*p* = 0.001) and FGF-2 (*p* = 0.024). E-Selectin (*p* < 0.001) and NRG-1 were significantly different at *p* = 0.026 at T2 compared to baseline. However, OPN and OSF-1 did not exhibit significant changes (*p* = 0.348; *p* = 0.065). Fontan patients show hypoxia-related protein patterns similar to healthy individuals despite intermittent hypoxemia, but their response to standardised hypoxia was described here for the first time, requiring further study.

## 1. Introduction

The diagnosis of a univentricular heart is based on numerous complex cardiac defects. In Germany, every year, ~200 children are born with a complex heart defect, leading to univentricular circulation [1].

The common pathway for these children ends with Fontan palliation as the procedure of choice today. This involves two to three surgical steps, which are usually performed within the first 5 years of life.

Until the final palliation, there are parallel circuits with a mixture of oxygen-poor and oxygen-rich blood, which leads to an average oxygen saturation of 75–85%. In order to separate the circulatory systems in the long term and thus enable full oxygenation of the blood, the superior and inferior vena cava are connected directly to the pulmonary artery without the interconnection of a subpulmonary ventricle [2] during the Glenn operation at the age of 6 months and the Fontan operation at the age of 3–5 years.

An oxygen saturation between 70 and 90% is generally very well tolerated by children in the first years of life and enables a largely normal life with tolerable limitations in physical capacity, whereas people who are not accustomed to such chronic hypoxemia would experience massive limitations. This is probably due to adaptation processes at the cellular level with more effective oxygen utilisation. After separation of the circuits, saturation increases to >90% in most cases. The investigation of hypoxia-induced factors and their significance in pathophysiological processes and adaptation reactions is becoming an increasingly prominent area of research [3], including in the field of cardiovascular medicine [4]. Particularly, it appears crucial to investigate these factors with respect to their influence on neogenesis of lymphatic vessels and angiogenesis, as these areas hold high relevance for disease progression in Fontan patients [5] who have experienced chronic hypoxia in their first years of life.

Hypoxia-inducible proteins such as VEGF play an important role in patients with a univentricular heart or a congenital heart defect, as has been shown in several studies with partly different results. Mori et al. described an increased level of VEGF in Fontan patients in association with the presence of aortopulmonary collaterals. The connection between the neogenesis of lymphatic vessels and VEGF was also demonstrated [6,7,8]. Suda et al. were able to show an increase in VEGF in children with cyanotic heart defects. Patients with biventricular correction showed a normalisation of VEGF, in contrast to univentricular palliated patients, in whom an increased VEGF was also measured [9]. In contrast, Haertel et al. demonstrated no difference in baseline values of adolescent and young adult patients after Fontan palliation and matched healthy controls [10].

Another relevant factor appears to be ANG2. It has been shown that elevated serum levels of ANG2 in Fontan patients serve as a risk factor for adverse events and mortality [11]. Finally, a significant relevance of interleukin-10 (IL-10) has been demonstrated in children with congenital heart defects and initial hypoxemia. In these cases, reduced IL-10 was observed [12]. Given that hypoxia reduces IL-10 [13] and IL-10 has the function of downregulating Vascular Endothelial Growth Factor (VEGF) [14], initially decreased IL-10 leads to increased expression of VEGF, contributing to the complications described earlier. Understanding the intricate relationships and signalling pathways involving these molecular effectors is essential not only for unravelling the molecular underpinnings of hypoxia but also for paving the way towards novel therapeutic interventions for patients with Fontan circulation. Therefore, it is of particular interest to understand the expression pattern of hypoxia-inducible proteins and effector molecules in people with chronic hypoxemia, as is the case in patients with univentricular circulation, especially before, but also after, Fontan palliation. Further, the effect of additional hypoxia due to, for example, staying in the mountains or air travelling, which is the subject of this research, is not yet sufficiently understood. To investigate this effect, blood samples from adolescent and young adult patients with Fontan palliation were analysed and compared under normal conditions and after 24 h hypoxia exposure including a stress test in simulated normobaric hypoxia equivalent to 2500 masl.

## 2. Results

### 2.1. Clinical Data

All participants tolerated hypoxia without relevant clinical symptoms. Nevertheless, an excessive drop in peripheral and capillary oxygen saturations could be measured, suggesting a cellular reaction. Capillary oxygen saturation was 83.9 ± 3.7% after 4 h of hypoxia exposure, 85.4 ± 4.1% before sleeping, and 84.7 ± 3.7% after awakening (*p* = 0.483). Daytime transcutaneous oxygen saturation (tcSpO_2_) was 92.5 ± 2.8% during normoxia and 86.2 ± 3.8% during hypoxia exposure (*p* < 0.001). None of the participants experienced oxygen desaturation < 75% overnight, which was the primary safety outcome. Several other values were measured over the study period and have been published elsewhere [15,16].

### 2.2. Hypoxia-Inducible Proteins and Effector Molecules

#### 2.2.1. IL-10

Study participants showed average baseline levels of interleukin-10 (IL-10) of 14.35 pg/mL with a standard deviation of ±1.87 pg/mL. After hypoxic exposure, IL-10 levels increased (mean 15.51 pg/mL ± 2.25 pg/mL, *p* = 0.001) (Figure 1A).

#### 2.2.2. CCL2

For CC chemokine ligand 2 (CCL2), the baseline mean was 118.2 pg/mL with a standard deviation of ±26.66 pg/mL. Hypoxia assessments revealed a heightened mean of 130.5 pg/mL ± 23.43 pg/mL, also demonstrating statistical significance (*p* = 0.006) (Figure 1B).

#### 2.2.3. VEGF

Vascular Endothelial Growth Factor (VEGF) at baseline displayed a mean of 13.18 pg/mL with a standard deviation of ±6.10 pg/mL. Subsequent to hypoxia, VEGF measurements resulted in a mean of 20.29 pg/mL ± 11.54 pg/mL, showing statistical significance (*p* = 0.014) (Figure 1C). Under normoxic (r = 0.071) and hypoxic (r = 0.099) conditions, no significant correlation was observed between tcSpO_2_ and VEGF. The correlation between the difference in tcSpO_2_ and VEGF between normoxic and hypoxic conditions was negative, with a correlation coefficient of r = −0.344 (Figure 2). There was no statistical significance in any of the three correlations.

#### 2.2.4. ANG-1

Angiopoietin-1 (ANG-1) at baseline demonstrated a mean of 4.31 ng/mL with a standard deviation of ±2.30 ng/mL. Following hypoxia, ANG-1 levels increased, resulting in a mean of 7.34 ng/mL ± 3.49 ng/mL, showing high statistical significance (*p* = 0.001) (Figure 1D).

#### 2.2.5. ANG-2

At baseline, Angiopoietin-2 (ANG-2) manifested a mean of 15.61 ng/mL ± 4.93 ng/mL. Following hypoxia, ANG-2 levels increased, resulting in a mean of 16.74 ng/mL with a standard deviation of ±5.20 ng/mL, indicating significance (*p* = 0.029) (Figure 1E).

#### 2.2.6. E-Selectin

E-Selectin at baseline displayed a mean of 23.43 ng/mL with a standard deviation of ± 7.21 ng/mL. Subsequent to hypoxia, the mean rose to 25.23 ng/mL with a standard deviation of ±7.63 ng/mL, demonstrating high statistical significance (*p* < 0.001) (Figure 1F).

#### 2.2.7. FGF-1

Fibroblast Growth Factor-1 (FGF-1) at baseline presented a mean of 104.3 pg/mL ± 22.44 pg/mL. After hypoxic exposure, FGF-1 levels increased, registering a mean of 111.5 pg/mL with a standard deviation of ±23.00 pg/mL, demonstrating statistical significance (*p* = 0.001) (Figure 1G).

#### 2.2.8. FGF-2

Fibroblast Growth Factor-2 (FGF-2) exhibited a baseline mean of 2.606 pg/mL with a standard deviation of ±0.78 pg/mL. After hypoxic exposure, FGF-2 levels increased, registering a mean of 2.940 pg/mL ± 1.05 pg/mL, demonstrating statistical significance (*p* = 0.024) (Figure 1H).

#### 2.2.9. NRG-1

Neuregulin-1 (NRG-1) at baseline had a mean of 102.6 pg/mL ± 24.81 pg/mL. Following hypoxia, NRG-1 levels increased, resulting in a mean of 106.6 pg/mL with a standard deviation of ±24.71 pg/mL, demonstrating significance (*p* = 0.026) (Figure 1I).

#### 2.2.10. Osteopotin

Osteopontin (OPN) baseline levels presented a mean of 46.97 ng/mL with a standard deviation of ±31.26 ng/mL, while hypoxia levels showed a mean of 47.98 ng/mL ± 30.05 ng/mL, though without statistical significance (*p* = 0.348) (Figure 1J).

#### 2.2.11. OSF-1

Periostin (OSF-1) at baseline featured a mean of 56.26 ng/mL with a standard deviation of ±20.31 ng/mL. Hypoxia measurements yielded a mean of 57.65 ng/mL with a standard deviation of ±20.80 ng/mL, lacking statistical significance (*p* = 0.065) (Figure 1K).

## 3. Discussion

This is the first study that describes the expression of hypoxia-induced molecules and effectors in the plasma of patients with Fontan circulation under normal circumstances and after exposure to simulated altitude under normobaric hypoxia (15.2% ambient oxygen content) > 24 h, corresponding to 2500 masl.

Isolated studies, which focus in particular on angiogenesis and lymphangiogenesis, describe the expression of individual proteins (especially VEGF and Angiopoietin 1 and 2) in patients with Fontan physiology [6,7,8,9]. A complex analysis of several factors has not yet been shown in connection with this hemodynamically special group. Due to the chronic hypoxemia of these patients, it can be assumed that different expression patterns are present compared to healthy control groups. Furthermore, it is of great interest to better understand the regulation of the individual molecules when exposed to an additional hypoxia stimulus and thus the clinical significance of this condition. It is still unclear why there are patients who, despite good hemodynamics, show massive collateral formation and pathological lymph distribution patterns even before Fontan completion, whereas others show no collateral formation at all. In this case, a purely mechanical explanation based on high resistance in the venous and pulmonary arterial system does not appear to be sufficient.

The baseline values of VEGF do not differ significantly to those from healthy individuals investigated in previous studies. Kraus and colleagues measured individual baseline levels between 15 pg/mL and 60 pg/mL in healthy volunteers before acute exercise [17].

Since the described cohort consists of adolescents and young adults who have had an oxygen saturation of >90% for at least 10 years under normal circumstances, it is conceivable that the expression patterns of the individual hypoxia-induced factors do not differ significantly from those of a normal cohort. This would mean that chronic hypoxemia in the first years of life after correction or palliation of the underlying heart defect does not lead to lifelong changes at the cellular level. An abundance of factors involved in immune response, the adhesion of immune cells to the endothelium, vasculogenesis, endothelial barrier function, and the proliferation and migration of endothelial cells were analysed.

IL-10 is known as a primarily anti-inflammatory cytokine released by monocytes or T-lymphocytes. IL-10 inhibits activated macrophages and the production of pro-inflammatory cytokines like IL-1β, TNFα or IFNγ, modulating the inflammatory response, as recently reviewed by Saraiva and colleagues [18]. The upregulation of IL-10 after hypoxic exposure may indicate a trend towards a reduced inflammatory reaction. However, the role of IL-10 in angiogenesis is discussed controversially, attributing both pro- and anti-angiogenic characteristics, especially in the context of tumour angiogenesis [19,20]. This controversy is also evident in the present study, which, in contrast to the data from Silvestre et al., observed a simultaneous increase in IL-10 and VEGF [14]. Whether the significant upregulation of IL-10 here correlates with increased or decreased angiogenesis needs further study, including long-lasting follow-up examinations. CCL-2 and E-Selectin are chemokines involved in the regulation of immune cell communication. To date, the effects of hypoxia on CCL-2 function has been intensively studied in pregnancy and embryonic development [21,22]. Hypoxic upregulation of CCL-2 could increase the adhesion of immune cells to the endothelium, which possibly results in an increased local inflammation reaction as well as the induction of angiogenesis. E-Selectin is exclusively expressed on endothelial cells, mediating the activation, adhesion, and transmigration of immune cells to the tissue. High levels of E-Selectin may be involved in an increase in endothelial inflammation [23], which could augment the adverse angiogenesis and lymphangiogenesis seen in patients with Fontan circulation. To precisely define the role of elevated CCL-2 and E-Selectin with respect to angiogenesis, long-term follow-up examinations are needed.

As hypoxia is a main driver of vasculogenesis, an expected significant increase in the release of VEGF, FGF1, and FGF2 could be measured here after the exposure of patients with Fontan circulation to normobaric hypoxic conditions comparable to an elevation of 2500 m asl. VEGF is the main driver of de novo vasculogenesis during embryogenesis [24]. It is known as one of the most important factors for angiogenesis and lymphangiogenesis, especially the growth of new blood and lymph vessels from existing vessels [25]. In this context, it has already been shown that VEGF normalises in children with cyanotic congenital heart failure after biventricular correction with subsequent normoxia. In contrast, this was not observed in Fontan patients despite normoxia having been achieved [9]. However, the described baseline values differ markedly within the working groups. Data demonstrated by Haertel et al. showed no significant differences under normal conditions for patients with Fontan physiology compared to matched controls (26.5 ± 24.6 pg/mL vs. 25.0 ± 22.2 pg/mL) [10]. The degree of increase under simulated hypoxic conditions, as illustrated in Figure 1C, has not been described by others and might be explained by the results of Mori et al., where the level of VEGF was dependent on the presence of aortopulmonary collaterals. Additionally, the factor of central venous pressure (CVP) appears to be a significant contributor, as an increase in CVP also results in an elevation in VEGF [8].

When these factors are considered collectively, it is probable that a combination of multiple influencing factors is responsible, although the primary trigger remains uncertain [9]. Analysis of the correlations between VEGF and tcSpO_2_ under normoxic and hypoxic conditions showed no significant direct relationship between the two parameters. However, when analysing the correlation of the differences between normoxia and hypoxia, a trend emerged. This observation suggests that the release of VEGF is not primarily influenced by the absolute saturation values, but rather by the extent of the decrease in oxygen saturation (Figure 2).

FGF1 and FGF2 belong to the FGF family, which possesses broad mitogenic and cell survival activities, and are involved in a variety of biological processes, including embryonic development, cell growth, morphogenesis, tissue repair, and tumour growth and invasion. These proteins function as modifiers of endothelial cell migration and proliferation, as well as angiogenic factors. In particular FGF1, FGF2, and the herein not analysed FGF4 show promising results in clinical trials to improve angiogenesis after myocardial infarction or angina pectoris [26]. The upregulation of FGF1 and FGF2 plasma levels in patients with Fontan circulation may therefore be an indicator of adverse collateral formation after hypoxic exposition.

The balance between ANG1 and ANG2 is important for the maintenance of the endothelial barrier as both molecules compete for the TIE-2 receptor expressed on endothelial cells [27]. An excess of ANG1 results in the stabilisation of the endothelial barrier, whereas excessive amounts of ANG2 are correlated with an increase in endothelial permeability. In healthy individuals, the concentration of ANG1 is about five times higher than the concentration of ANG2. However, in patients with Fontan circulation, the amount of ANG2 is three times higher than the amount of ANG1, so the balance between the two factors shifts to ANG2, which could be associated with a decrease in the endothelial barrier. This in turn could be a prerequisite for increased outgrowth of vessels. The more pronounced increase in plasma concentrations of ANG1 in comparison to an only moderate increase in ANG2 concentration may therefore not compensate for the increase in ANG2. Shirali et al. [28] described much lower values of Ang-2 in six Fontan patients under normal conditions and could not find a relationship between ANG2 and pulmonary arteriovenous malformations (PAVMs), suggesting that the role ANG2 plays in Fontan physiology is likely unrelated to PAVM formation. In our cohort, baseline levels of ANG-2 were nearly twice as high, and hypoxia exposure led to an additional increase. In the selection of patients for the present study with a saturation level of >90% at baseline, the almost doubled baseline values are not contradictory to the hypothesis of Shirali et al. that Ang-2 is not an indicator of PAVM or is correlated to other collateral formation. Moreover, since ANG1 was not measured in Shirali’s study, a final conclusion regarding the role of ANG2 in the formation of collaterals in patients with Fontan circulation could not be drawn.

Neuregulin-1 belongs to a family of four structurally related proteins that are part of the EGF family. These proteins play multiple essential roles in vertebrate embryogenesis including cardiac development. Particularly in diabetic cardiomyopathy and after ischemia, Neuregulin-1 is assumed to play a crucial role in myocardial blood supply by inducing coronary collateral formation [29]. Here, no significant upregulation of Neuregulin-1 after hypoxic exposure was observed, from which it can be concluded that Neuregulin-1 does not play a major role in collateral formation in patients with Fontan circulation.

Osteopontin is a ubiquitously expressed protein involved in the survival of lethally stressed cells. It has a chemotactic effect on macrophages, T cells, and dendritic cells. The lack of hypoxic induction is in a good agreement with the observed increase in the release of anti-inflammatory IL10, pointing towards a reduction in inflammation under hypoxic conditions.

Periostin (OSF-1) has been reported to be a critical player in the inflammatory microenvironment in various disorders such as airway inflammation, skin inflammation, atherosclerosis, and fibrosis [30]. Moreover, it is assumed to be involved in cardiac development. No increase in Periostin plasma concentration was detected in Fontan patients after hypoxic exposure.

The extent to which these hypoxia-inducible factors can be influenced by medication as a therapeutic option cannot be clarified in this study. The cohort is too small for a statement regarding cardiac insufficiency therapy. This must be mentioned as a limiting factor.

Another limitation of this study is the fact that patients were exposed to normobaric hypoxia for reasons related to patient safety and the feasibility of the study. The extent to which this influences the cellular hypoxia response in comparison to real altitude conditions (hypobaric hypoxia) has not yet been investigated and must be the subject of further studies.

As this is a pilot project, the data describe a small group size of Fontan patients with relatively high initial oxygen saturations > 90%. It is therefore not possible to transfer the data to all Fontan patients at this point.

## 4. Materials and Methods

### 4.1. Study Populations

A total of 18 adolescent and adult patients with Fontan circulation were included in the study. Inclusion criteria were stable circulatory conditions (NYHA I), peripheral oxygen saturation (SpO_2_) of >90%, and the ability to perform cardio-pulmonary exercise testing (CPET). Individuals with oxygen saturation < 90%, failing Fontan, Fontan tunnel fenestration, pulmonary artery stenting, pacemakers, medication with vasoactive drugs like sildenafil or bosentan, or claustrophobia and smokers were excluded. Apart from the heart defect, there were no other malformations. The following medication was taken: anticoagulation (*n* = 15), beta blockers (*n* = 5), and ACE inhibitors (*n* = 4).

All participants and, in the case of adolescents, their legal guardians gave written informed consent before they started with the investigation. The patient characteristics are presented in Table 1.

In accordance with the Declaration of Helsinki, the study protocol was approved by the Ethics Committee of the University of Bonn (application number 054/20), as well as by the North Rhine Medical Association (application number 2020046). The study protocol was registered at the German Clinical Trials Register (DRKS) (application number DRKS00025989).

### 4.2. Study Protocol and Sample Collection

In the period from April to May 2022, we investigated the influence of long-term normobaric hypoxia including nighttime sleep on patients with Fontan circulation. After several investigations under normal ambient conditions (normoxia), they were exposed to normobaric hypoxia in an altitude module (:envihab, DLR, Cologne, Germany) with ambient oxygen content of 15.2% equivalent to an altitude of 2500 masl. Therefore, oxygen was partially replaced by nitrogen (N_2_) at constant atmospheric pressure (≈1013 hPa). The total duration of the exposure was 24 h, including, among others, a CPET with submaximal exercise.

Venous blood samples were taken either via a central venous catheter (PIC), which was inserted via the brachial vein into the vena cava for pulmonary artery pressure measurements and was available for blood sampling throughout the study, or a peripheral venous line for those who did not consent to a PIC-line. The first sample was taken as a baseline under normoxic conditions before the first CPET was performed (T1). The second sample was obtained after the end of the CPET (submaximal step: protocol on recumbent ergometer bike after 24 h in hypoxia, T2) in order to capture the greatest possible hypoxia stimulus. The blood was collected into sodium heparin vacutainers (Becton, Dickinson and Company, Franklin Lakes, NJ, USA), centrifuged directly after sampling at 760× *g* for 15 min, and the supernatant was transferred into clean tubes and immediately stored at −80 °C.

### 4.3. SaO_2_ Measurement

Capillary oxygen saturation (SaO_2_) was measured at three timepoints after 4 h of hypoxia exposure, before sleeping and after awakening. The samples were taken from the patient’s fingertip using a lancing device and analysed in the ABL90 FLX Analyzer (Radiometer Medical, Copenhagen, Denmark).

### 4.4. SpO_2_ Measurement

SpO_2_ was measured several times during the day and continuously during nighttime and while performing CPET. Therefore, a finger pulse oximeter (IntelliVue X3, Philips, Amsterdam, The Netherlands) was used.

### 4.5. Sample Preparation

Venous blood was drawn prior to entering the altitude module in normoxia and in the phase of maximal hypoxia. The coagulated samples were centrifuged (760× *g* for 15 min), and serum aliquots were stored at −80 °C for subsequent analyses. In serum samples, the following 10 parameters were assessed using custom-made Luminex™ multiplex arrays (R&D Systems, Minneapolis, MN, USA): IL-10, CC-chemokine Ligand 2 (CCL2), Angiopoietin-1 (ANGPT1), Angiopoietin-2 (ANGPT2), fibroblast growth factor acidic (FGF-1), Fibroblast Growth Factor 2 (FGF-2), Neuregulin-1 beta 1 (NRG-1-beta-1), Osteopoetin (OPN), Periostin (OSF-1), E-selectin (CD62E), and Vascular Endothelial Growth Factor (VEGF).

All analyses were performed according to the manufacturer’s protocol. Bead-based multiplex arrays such as the Luminex™ system are described in the work from Zhang et al. [31]. Arrays were analysed using a FlexMap™ reader (Luminex Corp., Austin, TX, USA). Results are given in pg/mL serum.

### 4.6. Statistical Analysis

Data are presented as mean ± SD. Statistical analysis was performed with GraphPad Prism 10.1.0. The Shapiro–Wilk test was used to analyse whether a normal distribution was present. In the presence of a normal distribution, the statistical analysis was performed using a paired *t*-test. When values did not show a normal distribution, the data were analysed using the Wilcoxon test. Correlation coefficients (r) ≥ 0.7 and ≤−0.7 were assumed to indicate a correlation between these variables. Statistical significance was accepted at *p* ≤ 0.05.

## 5. Conclusions

The expression patterns of hypoxia-inducible proteins and effector molecules of patients with univentricular circulation after Fontan palliation correspond to those of healthy comparison collectives under normal conditions despite intermittent hypoxemia due to physical activity and exercise. The cellular response to a maximum hypoxemia stimulus by exposing the patients to normalised hypoxia, equivalent to 2500 masl with additional physical stress, was described here for the first time, so a comparison with previous studies is not possible. The impact of these patterns must be considered in the context of a massive change in circulatory conditions in the case of Fontan circulation. The extent to which the response was influenced by the application of normobaric hypoxia instead of real hypobaric hypoxia must be part of further investigations. The transfer of data into clinical contexts is an important aspect for the future. The extent to which therapeutic consequences can be derived from this, e.g., on the pathologies of the lymphatic system or the sometimes extensive formation of collaterals, must be investigated in further studies. In particular, the development of the individual parameters in the first years of life in the context of gradual surgical palliation up to Fontan surgery is of great interest here, as the hypoxemic conditions change fundamentally during this period. Whether hypoxemia present in the first years of life with arterial oxygen saturation values between 70 and 90% in patients with univentricular circulation has a favourable, or any, effect at all on the cellular response to hypoxia exposure later in life must be the subject of further projects.

## Figures and Tables

**Figure 1 ijms-26-00887-f001:**
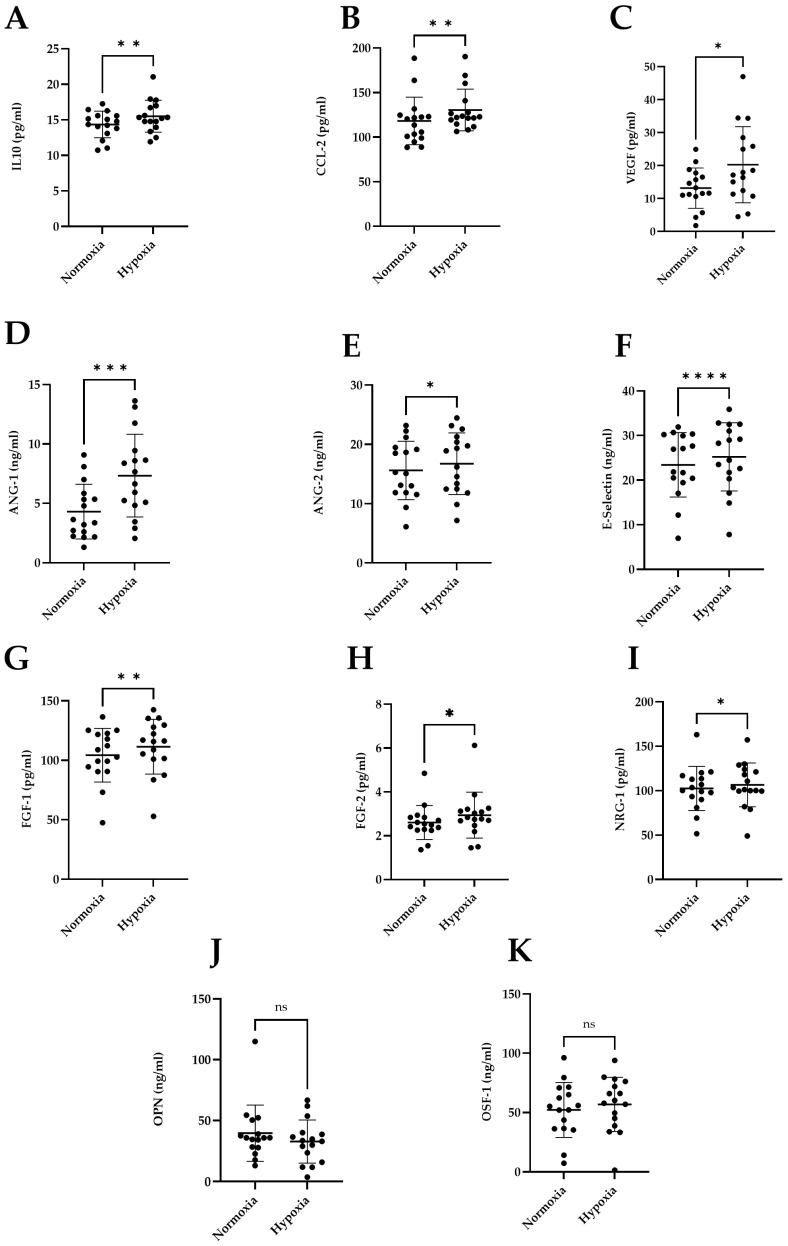
Analysis of hypoxia-inducible proteins and effector molecules in Fontan patients under normoxia and hypoxia given in mean ± SD: (**A**) interleukin-10 (IL-10); (**B**) CC-chemokine ligand 2 (CCL2), (**C**) Vascular Endothelial Growth Factor (VEGF), (**D**) Angiopoietin-1 (ANG1), (**E**) Angiopoietin-2 (ANG2), (**F**) E-selectin (CD62E), (**G**) fibroblast growth factor acidic (FGF-1), (**H**) fibroblast growth factor 2 (FGF-2), (**I**) Neuregulin-1 beta 1 (NRG-1-beta-1), (**J**) Osteopontin (OPN), (**K**) Periostin (OSF-1). * = ≤0.05, ** = ≤0.01, *** = ≤0.001, **** = ≤0.0001, ns > 0.05.

**Figure 2 ijms-26-00887-f002:**
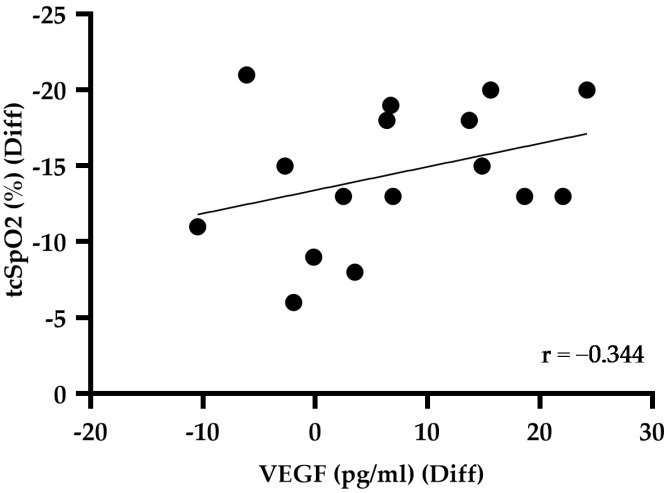
Correlation between the difference in transcutaneous oxygen saturation (tcSpO_2_) and Vascular Endothelial Growth Factor (VEGF) levels in normoxic and hypoxic conditions (r = −0.344).

**Table 1 ijms-26-00887-t001:** Patient characteristics.

Age (median [range])	Years	24.5 [16.3–38.8]
Gender	f/m	9/9
SpO_2_ (mean ± SD)	Normoxia	94.2 ± 3.1
SpO_2_ (mean ± SD)	Hypoxia	79.7 ± 6.0
Congenital heart malformation		
Tricuspid Atresia (TA)	4	
Pulmonary Atresia (PA)	1	
PA with congenitally corrected transposition of the great arteries (ccTGA)	2	
Hypoplastic Left Heart Syndrome (HLHS)	2	
Double-Inlet Left Ventricle (DILV)	5	
Double-Outlet Right Ventricle	2	
Double-Inlet Right Ventricle	1	
Single Ventricle with ccTGA	1	
Systemic Ventricle		
Left	10	
Right	8	

## Data Availability

Data is contained within the article.

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
