# Peer review of "Influence of Simulated Altitude Exposure (2500 m) on Patients with Fontan Palliation Based on Circulating Hypoxia-Associated Factors"

_ijms, 2025, doi:10.3390/ijms26030887_

Round 1
Reviewer 1 Report
Comments and Suggestions for Authors
In this study, sixteen patients who previously underwent a Fontan procedure or congenital heart disease were subjected to hypoxic conditions (15% FiO2) for 24 hours. the authors report modest changes in circulating levels of angiogenic and inflammatory proteins, similar to what is observed in healthy control volunteers. The study has clinical relevance, since many patients undergo the Fontan procedure. the study is well conducted but i have several comments.
1) One omission is the lack of a control group. ie, patient who did not undergo a surgical correction for congenital heart disease. the authors reference other work, but it would markedly improve the impact of the study and the paper if this data were included. perhaps all patients, regardless of their cardiac condition, have similar changes to hypoxia. is this information available?
2) the authors measured various proteins involved in angiogenesis (growth factors) and inflammation. however, some of the standard inflammatory cytokines such as IL-2, IL-8 and TNF-alpha were not examined. The anti-inflammatory cytokine IL-10 was examined. do the authors have the additional data or is blood available to do this additional analysis?
Author Response
Comments 1: One omission is the lack of a control group. ie, patient who did not undergo a surgical correction for congenital heart disease. the authors reference other work, but it would markedly improve the impact of the study and the paper if this data were included. perhaps all patients, regardless of their cardiac condition, have similar changes to hypoxia. is this information available?
Dear Reviewer, thank you for the opportunity to improve our manuscript. Here are the responses to your questions/suggestions:
Response 1: We totaly agree with you. The comparison of a control group could have significantly enhanced the present study, but unfortunately this was not practically feasible due to the complex study design with a total study duration of 4 days per patient . For this reason, we did not include a control group. Therefore no changes were mede in the manuscript.
Comments 2: The authors measured various proteins involved in angiogenesis (growth factors) and inflammation. however, some of the standard inflammatory cytokines such as IL-2, IL-8 and TNF-alpha were not examined. The anti-inflammatory cytokine IL-10 was examined. do the authors have the additional data or is blood available to do this additional analysis?
Response 2: None of the patients displayed signs of inflammation before exposure to hypoxia. Measurement of pro-inflammatory cytokines like IL-1b or TNFα under control conditions or in healthy individuals is most difficult, since the plasma concentrations are below the detection limit of commercial available ELISAs or Luminex assays. Because the expression maximum of this cytokines usually occurs after about 3 to 6 hours and blood samples were collected after 24h of hypoxic exposure, we decided to forego the measurement of this typical pro-inflammatory cytokines in our Luminex assay.
Our decision is supported by the work of Michel and coworkers, who performed proteomic analyses in patients with Fontan circulation in comparison to healthy controls. No differences in expression of IL-1b, TNFα or IL-8, but significant changes in the expression of syndecan-1, glycophorin-A, leukemia inhibiting factor and nerve growth factor were described [1]. However, whether plasma concentrations of these factors are influenced by hypoxic exposure needs further analyses. Unfortunately, blood for this additional analyses is no longer available.
[1] Michel M, Renaud D, Schmidt R, Einkemmer M, Laser LV, Michel E, Dubowy KO, Karall D, Laser KT, Scholl-Bürgi S. Altered Serum Proteins Suggest Inflammation, Fibrogenesis and Angiogenesis in Adult Patients with a Fontan Circulation. Int J Mol Sci. 2024 May 16;25(10):5416. doi: 10.3390/ijms25105416. PMID: 38791454
We hope that we have answered your questions satisfactorily. If this is not the case, please contact us again.
Reviewer 2 Report
Comments and Suggestions for Authors
The article entitled "Influence of simulated altitude exposure (2500 m) on patients with Fontan palliation on circulating hypoxia-associated factors" is interesting as it describes a new procedure and its results in a specific group of patients. However, I would have some suggestions:
1. First of all, the structure of the manuscript is not correct. After introduction, the correct order is - material and methods (with all subtypes) including statistical analysis, after that results, discussion and conclusion.
2. Which are the exclusion criterias?
3. The age range is between 16 and 39 years. Could this influence the results? A patient of 16 years has a shorter expose the pathology and its influence of other systems than a patient of 39 years.
4. Did the patients have other associated pathologies/malformations?
5. What treatment did the patients have? Could it possibly influence the results?
6. Are there any other molecules in the literature regarding aspect which was studies?
7. It would be interesting to discuss about other therapeutical approach since no one until now analysed your aspect. After that, I would make a comparation and I would discuss about the importance of this analysis in future directions.
Author Response
Dear Reviewer, thank you for your critical and constructive review of our manuscript. We think that the manuscript has greatly profited from your review and we have tried to respond to all of your comments point by point and have changed the manuscript accordingly.
Comment 1: First of all, the structure of the manuscript is not correct. After introduction, the correct order is - material and methods (with all subtypes) including statistical analysis, after that results, discussion and conclusion.
Response 1: The structural organisation was based on the specifications of the journal MDPI, therefore no changes were done.
Comments 2: Which are the exclusion criterias?
Response 2: The exclusion criteria are listed on page 9, lines 317-319. "Individuals with oxygen saturation <90%, failing Fontan, Fontan-tunnel fenestration, pulmonary artery stenting, pacemakers, medication with vasoactive drugs like sildenafil or bosentan, claustrophobia, or smokers were excluded. " For this reason, no changes have been made
Comments 3: The age range is between 16 and 39 years. Could this influence the results? A patient of 16 years has a shorter expose the pathology and its influence of other systems than a patient of 39 years.
Response 3: Based on the selection of patients with stable haemodynamics and oxygensaturation values >90%, an influence of age is not to be expected in this patient collective. A primary effect of age is more likely to be expected in the first years of life prior to Fontan completion. Since oxygensaturation and haemodynamics change substantially here.
Comments 4: Did the patients have other associated pathologies/malformations?
Response 4: Thank you for this question. The Fontan patients assessed here had no other pathologies or malformations that could have had an impact on the results. For better understanding, this has been changed in the manuscript in the exclusion criteria section.
"Apart from the heart defect, there were no other malformations."
Change can be found Page 9 line 319-320.
Comments 5: What treatment did the patients have? Could it possibly influence the results?
Response 5: Thank you for this constructive advice. We have adjusted the text in this regard on page 8 line 298-301 page 9, lines 320-321.
"The following medication was taken: Anticoagulation (n=15), beta blockers (n=5) and ACE inhibitors (n=4)."
Comments 6: Are there any other molecules in the literature regarding aspect which was studies?
Response 6: Most of the studies addressing the effects of hypoxia in patients with Fontan circulation focussed on systemic parameters like oxygen uptake, right ventricular pressure and cardiac function [2]. The abundance and regulation of plasma biomarkers in Fontan patients after exposure to hypoxia is mainly unknown. Based on the proteomic data published by Michel et al. [1, herein supplemental table 1] molecules could be evaluated with respect to changes in plasma concentration after hypoxia. No change were done.
[2] Staempfli R, Schmid JP, Schenker S, Eser P, Trachsel LD, Deluigi C, Wustmann K, Thomet C, Greutmann M, Tobler D, Stambach D, Wilhelm M, Schwerzmann M. Cardiopulmonary adaptation to short-term high altitude exposure in adult Fontan patients. Heart. 2016 Aug 15;102(16):1296-301. doi: 10.1136/heartjnl-2016-309682. Epub 2016 May 23.
Comments 7: It would be interesting to discuss about other therapeutical approach since no one until now analysed your aspect. After that, I would make a comparation and I would discuss about the importance of this analysis in future directions.
Response 7: We agree with this comment. The conclusion was therefore modified. This change can be found page 11 line 396-402. "The transfer of data into clinical contexts is an important aspect for the future. The extent to which therapeutic consequences can be derived from this, e.g. on the pathologies of the lymphatic system or the sometimes extensive formation of collaterals, must be investigated in further studies. In particular, the development of the individual parameters in the first years of life in the context of gradual surgical palliation up to Fontan surgery is of great interest here, as the hypoxaemic conditions change fundamentally during this period."
We would also like to mention that we are currently in the process of carrying out further studies on patients with univentricular heart before undergoing Fontan completion. In our opinion, the greatest effects can be expected during this period of hypoxaemia.
We hope that we have answered your questions satisfactorily. If this is not the case, please contact us again.
Reviewer 3 Report
Comments and Suggestions for Authors
The submitted manuscript entitled “Influence of stimulated altitude exposure (2500 m) on patients with Fontan palliation on circulating hypoxia-associated factors” reports how exposure to hypoxia can have an impact on patients with univentricular heart live with chronic hypoxia.
There are not many studies that deal with this type of research. The topic is original.
The aim of this study was therefore to investigate the relationships between the effects of additional hypoxia, e.g. due to air travel or stays in the mountains, in patients with a univentricular circulation. The results obtained show that there are significant changes in the expression of hypoxia-induced molecules and effectors in patients with Fontan palliation. The results shown in two figures are clear and nicely.
The authors used an appropriate methodology.
The conclusions are consistent with presented evidence and arguments.
One limitation of this study is the small group size of Fontan patients. Therefore, it would be good to conduct further in vivo research on suitable animal models.
Author Response
Comments 1: One limitation of this study is the small group size of Fontan patients. Therefore, it would be good to conduct further in vivo research on suitable animal models.
Response 1: Many thanks for the positive feedback on our work. We agree with you regarding the limited number of patients. Because of the complex study design and the long study period of 4 days/patient, it was unfortunately not possible to recruit more patients. However, we are currently planning further studies, particularly on children with univentricular hearts who have not yet undergone Fontan completion. In addition, there is a working group at our centre that has developed an animal model in pigs for studies on univentricular hearts, so that we are also planning further studies on hypoxia in this field.
Round 2
Reviewer 2 Report
Comments and Suggestions for Authors
The authors repsponded to all of my questions. The article is suitable for publication.